# Teicoplanin-Resistant Coagulase-Negative Staphylococci: Do the Current Susceptibility Testing Methods Reliably Detect This Elusive Phenotype?

**DOI:** 10.3390/antibiotics12030611

**Published:** 2023-03-19

**Authors:** Adriana D. Balasiu, Colin R. MacKenzie

**Affiliations:** Institute of Medical Microbiology and Hospital Hygiene, Heinrich-Heine University Hospital, 40225 Dusseldorf, Germany

**Keywords:** CoNS, teicoplanin, therapy, resistance, susceptibility testing

## Abstract

Coagulase-negative staphylococci (CoNS), members of the skin commensal microbiota, are increasingly associated with local or systemic infections due to a shift in patient populations in recent decades. Subsequently, more CoNS strains have been subjected to antibiotic susceptibility testing (AST), thus leading to the increased detection of teicoplanin resistance. However, data concerning teicoplanin resistance among CoNS strains remain limited, heterogeneous, and inconclusive. We collected 162 consecutive CoNS strains identified using Vitek-2 as teicoplanin-resistant and tested them with a range of AST methods. The results of standard and high inoculum broth microdilution (sBMD; hBMD), agar dilution (AD) after 24 h and 48 h incubation, standard and macrogradient diffusion strip (sGDT, MET), screening agar, and disc diffusion were compared to assess their robustness and to establish a diagnostic algorithm to detect teicoplanin resistance. sBMD was used as the reference method, and the lowest number of strains were teicoplanin-resistant using this method. sGDT and disc diffusion generated similar results to sBMD. Compared with sBMD, AD-24 h generated the lowest number of false teicoplanin-resistant strains, followed by hBMD, AD-48 h, and Vitek-2. sGDT, a fast, easy, affordable method in diagnostic settings, generated the highest rate of false teicoplanin-susceptible strains. Vitek-2 testing produced the highest number of teicoplanin-resistant strains. Only in two strains was the initial Vitek-2 teicoplanin resistance confirmed using five other AST methods. In conclusion, the different antibiotic susceptibility testing methods generated inconsistent, inconclusive, and discrepant results, thus making it difficult to establish a diagnostic algorithm for suspected teicoplanin resistance. Teicoplanin testing proved to be challenging and easily influenced by technical factors. This study aimed not only to raise awareness of teicoplanin resistance testing but also of the need for future studies focusing on the clinical efficacy of teicoplanin in relation to its susceptibility results.

## 1. Introduction

The coagulase-negative staphylococci (CoNS) include a large number of different *Staphylococcus* species and are part of the skin and mucous membrane commensal microbiota. In certain circumstances (interference with skin health, ecology, and structure, or the immune system), they may cause opportunistic local or systemic infections. Advances in modern medicine have led to an increased role of CoNS among patients who are immunocompromised, critically ill, long-term hospitalized, or have implanted medical devices [1,2,3,4,5,6].

CoNS strains have been reported to play a significant role not only among device-associated infections (intravascular catheters, cerebrospinal fluid shunts, prosthetic joint, vascular grafts, and peritoneal dialysis catheters) but also in osteomyelitis, infective endocarditis [3], surgical site infections [5], and infections in neonates [7]. Van Epps et al. showed [1] that 50–70% of healthcare-associated infections in the USA are a consequence of a broad spectrum of available implantable medical devices, from the easily replaceable peripheral cannula to long-term devices, including extracorporeal life support, left ventricular assist devices, neurological devices, and joint prostheses.

CoNS strains cause 20–30%, and in some studies even up to 45% [2,8] of central-line-associated bloodstream infections (CLABSIs) in intensive care units and 35–55% of cardiovascular implantable electronic device (CIED) infections [9]. Furthermore, the 2018 ECDC report showed that [5], overall, 50% of surgical site infections (SSIs) are due to Gram-positive cocci. CoNS strains were found in 26.4% of SSI after coronary artery bypass graft and 18.9% after hip prosthesis surgery. Amat-Santos et al. found that 24.5% of prosthetic valve endocarditis cases after transcatheter aortic valve replacement were caused by CoNS [3]. In addition, CoNS is a major cause of late-onset sepsis among neonates [7].

Different AST methods, depending on the setting, can be performed: semi-automated or manually, using microdilution or agar dilution, disc diffusion, or gradient test. Most CoNS strains display resistance to beta-lactam agents; therefore, glycopeptide antibiotics (GAs) are often the therapy of choice for these infections. Vancomycin and teicoplanin are naturally occurring actinomycete-derived glycopeptide antibiotics [10]. GAs share the same mechanism of action (inhibition of the cell synthesis), structure, and spectrum of activity (mainly aerobic Gram-positive bacteria). GAs bind to the N-Acyl-D-Ala-D-Ala subunit of peptidoglycan, thus inhibiting cell-wall biosynthesis and inducing cell death [11]. Teicoplanin has similar efficacy to vancomycin but has been associated with fewer side effects and less nephrotoxicity than vancomycin [12,13]. Therefore, teicoplanin has become a therapeutic alternative to vancomycin for certain patients (e.g., those with neutropenia [14], or renal dysfunction). 

Teicoplanin resistance has been increasingly reported over the years, but the published results are disparate. In our laboratory, we have made the same observation, and thus the main concern as to whether teicoplanin resistance is increasing remains unanswered. This leads to the question of which method is the most reliable to detect resistance to ensure that patients receive the appropriate therapy. Therefore, the aims of this study were to (i) assess the robustness of the routinely employed susceptibility testing by comparing it with other available methods and (ii) propose a diagnostic algorithm to detect the teicoplanin resistance and heteroresistance, thus avoiding labor-intensive population analysis methods.

## 2. Results

### 2.1. Patients and Included Isolates

Of the 162 tested isolates, 157 (96.9%) were *Staphylococcus epidermidis*, followed by *S. hominis* (3 isolates, 1.9%) and *S. haemolyticus* (2 isolates, 1.2%). In total, 96 (59.2%) strains were recovered from blood cultures, 76 (46.9%) of which were peripheral, and 20 (12.3%) were from central lines. The remaining 66 strains (40.8%) were isolated from tissue, intraoperative swabs, catheter tips, cerebrospinal fluid (CSF) from external ventricular drains (EVD), aspirates, respiratory samples, urine (from immunocompromised patients), and cell culture media (cell therapy products).

### 2.2. Vitek-2

The number of teicoplanin-resistant strains detected using Vitek-2 in our laboratory varied over 6 years between 20% and 32%, as shown by the annual resistance statistics listed in Table 1.

On retesting the 162 isolates using Vitek-2, 88 (54.3%) strains were susceptible to teicoplanin, and 74/162 (45.7%) were resistant. Most of the teicoplanin-resistant strains found with Vitek-2, i.e., 46/74 (62.2%), had a MIC of 8, while 27/74 (36.5%) had a MIC of 16, and 1 strain (1.3%) an MIC of 32. Moreover, Vitek-2 MIC distribution shows that the MICs are within close range of EUCAST defined teicoplanin breakpoint (Table 2 and Table 3). Vitek-2 found 63 teicoplanin-resistant strains not confirmed by sBMD, and in 14 strains, sBMD testing correlated with the Vitek-2 results (Table 2).

Using Vitek-2, 79% of the strains were oxacillin-resistant, and none of the strains displayed resistance to vancomycin or linezolid. All the antibiotics tested using Vitek-2 are listed in Table 3.

### 2.3. Standard and High-Broth Microdilution (sBMD and hBMD)

Using sBMD, 146/162 (90.1%) were teicoplanin-susceptible, and 16/162 (9.9%) were resistant. With hBMD, only 109/162 strains (67.3%) were susceptible, and 53/162 (32.7%) were resistant. With sBMD, most of the strains (62.9%) had a MIC of 2 or 4, whereas using hBMD, the majority (58.0%) had a MIC of 4 or 8. The 39 (24.1%) teicoplanin-resistant strains in the hBMD assay had a MIC of 4 (28 strains), 2 (10 strains), and 0.5 (1 strain) in sBMD. These results are summarized in Table 4, Table 5 and Table 6.

Vancomycin MIC was measured by means of sBMD and Vitek-2. All the samples were vancomycin-susceptible using both methods. While with sBMD, the majority 144/162 (88.9%) of the strains had a MIC of 2 mg/L, with Vitek-2, 70 (43.2%) had a MIC of 1 mg/L and 80 (49.4%) had a MIC of 2 mg/L.

### 2.4. Agar-Diffusion 24 h and 48 h Incubation (AD-24 h and AD-48 h)

In the AD-24 h assay, 128 (79%) strains were teicoplanin-susceptible, 33 (20.4%) were resistant, and 1 strain displayed no growth after 20–24 h incubation. Among the 33 teicoplanin-resistant strains in hBMD, 20 (12.3%) strains were susceptible, and 13 were resistant using sBMD, while with Vitek-2, 7 were susceptible, and 26 were resistant. AD-24 h, on the one hand, failed to recognize accurately 3 teicoplanin-resistant strains from sBMD, but on the other hand, generated 20 more resistant strains than sBMD. AD-24 h and the other AST results are summarized in Table 7 and Table 8.

In AD after 48 h incubation, only 98 (60.5%) strains remained susceptible, 63 (38.9%) were resistant, and 1 strain displayed no growth. Notably, 31 strains, initially tested in AD-24 h as susceptible with a MIC of 4 (30) and 2 (1), were resistant after 48 h incubation. Only 14 of the 16 teicoplanin-resistant strains in sBMD were among the 63 teicoplanin-resistant strains in AD-48 h. Testing with sBMD and AD-48 h found the highest number of susceptible strains, whereas using Vitek-2 and AD-48 h, most strains were teicoplanin-resistant (51). Further results are depicted in Table 9.

### 2.5. Standard Gradient Diffusion Test (sGDT) and Macrodilution Gradient Test (MET)

All but three strains tested teicoplanin-susceptible by means of sGDT. Most of the strains displayed a MIC of 1 mg/L (81/162) or 2 mg/L (44/162). The assay recognized only 3 of the 16 teicoplanin-resistant strains from sBMD, thus generating the highest rate not only of CA but also of vME (Table 10).

The values obtained using MET are not strictly speaking MICs. After 48 h incubation, 157/162 (96.9%) strains displayed growth at a MIC lower than 8 mg/L, 1 strain at 8 mg/L, 1 strain at 12 mg/L, and 2 (1.2%) (both *S. epidermidis*) strains failed to grow. The strains displaying growth at 8 mg/L were also tested for vancomycin resistance because according to the EUCAST criteria, the reading of teicoplanin at 8 mg/L is not enough in itself to assign a strain as vancomycin-resistant or as a heteroresistant strain. The two strains with high MET readings were confirmed using all other AST assays, except via AD-24 h and disc diffusion. The AST results are collated in Table 11.

### 2.6. Disc Diffusion and Screening Agar

By means of disc diffusion, all the samples except two were susceptible, according to the CLSI criteria, thus confirming that this method does not reliably detect teicoplanin resistance.

By means of screening agar (5 mg/L teicoplanin) using a standard 0.5 McF inoculum, 113/162 (69.8%) strains were positive, suggesting a teicoplanin MIC of over 5 mg/L and thus resistant. The remaining 44 (27.2%) were negative, 4 could not be evaluated, and 1 was not performed. Notably, 147/162 (90.7%) strains were positive when an McF 2 inoculum was used, 12 were negative, 2 could not be evaluated, and in 1, this was not performed. The most positive strains in a screening method, 99 (61.1%) strains with McF 0.5 and 131 (80.9%) using McF 2, were among the strains tested susceptible with sBMD and therefore would be falsely assigned as teicoplanin-resistant, which would correspond to the highest ME among all the employed AST methods. A summary of the results comparing the AST methods is depicted in Table 12.

## 3. Discussion

The AST results and the institutional yearly resistance statistics confirmed the previously published data [4,15,16,17] that CoNS strains are highly resistant to most commonly used beta-lactam antibiotic agents, leaving glycopeptides, linezolid, and daptomycin as the most important therapeutic options. A number of aspects should be considered when choosing the appropriate treatment, including side effects, risk of developing resistance during therapy, therapeutic drug monitoring, cost, and availability. Teicoplanin has been considered an alternative to vancomycin due to its lower nephrotoxicity, reduced drug interactions, and once-daily administration. 

Teicoplanin resistance has been reported in the USA and the UK since the early 1980s, but the published data since then [18] do not reflect the actual incidence and its impact on therapeutical use. Teicoplanin resistance is an increasing and emerging challenge, but published data are inconclusive due to a number of factors. These include the different methods employed (e.g., broth microdilution vs. disk diffusion [4]); settings, diagnostic vs. research (e.g., broth microdilution vs. population analysis); the standards employed (e.g., CLSI vs. EUCAST defined breakpoints); the inclusion of diverse cohorts (e.g., catheter-related bacteremia vs. healthy volunteers [17,19]; the bacterial species studied (most studies have focused on *S. aureus* and fewer on CoNS [20]); clonal dissemination [21]; data generated at different time points [22]; or that teicoplanin was not tested. Thus, to date, reports have probably underestimated the true incidence of teicoplanin resistance and are still insufficient to identify its underlying mechanisms with certainty. 

It is still unclear if increasing teicoplanin resistance should be attributed to one or several possible underlying mechanisms. The mechanism is neither well defined nor adequately studied. Several mechanisms have been proposed such as cellular aggregates and antibiotic retention [23] or cell-wall alteration through reorganization or thickening [24,25]. Perhaps even more worrying is that teicoplanin resistance has been shown to develop under therapy [26,27]. Biavasco et al. pointed out that the AST employed for teicoplanin can be easily influenced by technical factors such as methods, media, inoculum, and incubation time [28]. Furthermore, it has been shown that the physical properties of teicoplanin—a large, lipophilic, and negatively charged molecule—have an impact upon testing by generating a lower diffusion coefficient on agar compared with vancomycin [29]. 

Broth microdilution is generally regarded as the gold standard method for antibiotic susceptibility testing; however, few laboratories use it for routine purposes. To optimize laboratory workflow with a high sample throughput, semi-automated devices such as Vitek-2 are employed routinely for the AST of fast-growing bacteria. Generally speaking, Vitek-2 performs well: It is fast and robust, with minimum hands-on time, is cost-effective, and requires little technical expertise. In our laboratory, using Vitek-2, a rapid rise in teicoplanin-resistant CoNS strains was observed in 2015. Baris et al. also reported an increased number of teicoplanin-resistant strains with BD Phoenix [16]. As in our study, most of the samples tested as teicoplanin-resistant using Vitek-2 were not confirmed via sBMD, leading to the highest rate of ME among the AST methods. The majority of teicoplanin MIC, either 4 or 8 mg/L (56.8%), determined using Vitek-2 were close to the EUCAST epidemiological cut-off (ECOFF) value for CoNS (MIC 4 mg/L), thus having an impact upon the generated EA and CA. Meanwhile, most of the MIC in sBMD (54.7%) were concentrated at the upper limit of the range (2 or 4 mg/L), thus conforming to the published EUCAST MIC distribution determining the teicoplanin breakpoints for CoNS. Vaudaux et al. found a similar MIC distribution using macrodilution but not microdilution. Moreover, MIC distribution was different when performed using macrodilution or microdilution [30]. 

According to these results, the AST performance for teicoplanin does not fulfill the CLSI criteria of 90% agreement for both EA and CA [31]. It is difficult to establish a diagnostic workflow that reliably confirms teicoplanin resistance among routinely tested strains. Firstly, EA and CA differ in test, antibiotic, and methodology, confirming the results of Campana et al. Moreover, their results showed that EA and CA vary with species (e.g., CA for strip test for *S. aureus* (100%) vs. 75% for CoNS according to CLSI) [32]. Secondly, most of the routinely employed AST assays use a low bacterial inoculum and are fast, whereas the strains that might bear heteroresistance are first detected at CFU above 10^6^ CFU/mL and after a longer incubation time (48 h). With a final inoculum of 5 × 10^5^ CFU/well, the microdilution assay, the current gold standard method, is unable to reliably detect heteroresistance [30]. Routinely employed methods probably do not detect heteroresistant strains, which may have a negative impact on therapeutic outcomes. 

Teicoplanin AST is easily influenced by inoculum, incubation time, media, and method and is more variable than vancomycin AST. All these suggest that a re-evaluation of diagnostic methods, breakpoints, and their capacity to accurately identify teicoplanin resistance and heteroresistance among clinically relevant CoNS strains is needed. A possible diagnostic algorithm should encompass different steps that can be carried out in a routine setting: a rapid automated AST to identify possible resistance, followed by a high inoculum and a longer incubation period method to confirm resistance or susceptibility. The second method should preferably be fast, commercially available for routine settings, have a low cost, and be reliable and reproducible. A possible option would be MET. MET is a method with low hands-on time, but adjustments are needed for it to be as reliable for use with CoNS as it is for *S. aureus*. The strains with suspected teicoplanin resistance could be further tested in reference laboratories by means of population analysis profiles (PAPs). PAP is the gold standard method to detect heteroresistance. This is a demanding time-consuming method, difficult to implement in a routine setting, and poses the risk of selecting resistance instead of finding it [33]. Using a different method in the second step is challenging because not all laboratories have the option to produce the necessary in-house plates.

These results do not confirm an increased vancomycin resistance as previously thought or predicted. This may be due to an underlying mechanism that involves only teicoplanin or that the teicoplanin molecule presents technical difficulties causing an unreliable result [3]. A similar situation applies to colistin [34].

In conclusion, extensive teicoplanin susceptibility testing showed that the results obtained using a single method could not be fully confirmed by employing various other methods. Due to a high discrepancy among the methods tested, no algorithm can be proposed to reliably detect teicoplanin resistance. The fact that the results were so diverse suggests that all the aspects involved in teicoplanin testing should be re-evaluated so that improvements can be made not only in the laboratory but also in establishing reliable breakpoints. Given the relevance that these results pose for antibiotic therapy, further clinical studies looking into the clinical efficacy of teicoplanin and in vitro teicoplanin testing are of great importance. 

## 4. Materials and Methods

In accordance with the Declaration of Helsinki (2013), ethical approval for this study was granted by the Ethics Committee of the Medical Faculty of Heinrich Heine University, Dusseldorf (Study No. 5694/26.9.2016).

### 4.1. Bacterial Strains

For this study, 162 consecutive CoNS strains were collected from August 2015 to August 2016 at the Institute of Medical Microbiology and Hospital Hygiene, Heinrich Heine University Hospital, Düsseldorf. The strains were selected based on non-susceptibility against teicoplanin and were recovered from different samples such as blood culture, soft-tissue infections, or central lines. Routinely, putative clinically relevant isolates were subjected to identification and susceptibility testing. Identification was performed with Vitek^®^ MS (bioMérieux, Marcy l’Etoile, France), a matrix-assisted laser desorption ionization–time of flight mass spectrometry method (MALDI-TOF MS). Antibiotic susceptibility testing was performed with Vitek 2 (bioMérieux, Marcy l’Etoile, France) AST- P654 cards. 

The strains were stored in 80% glycerol in a Mueller–Hinton Broth (MHB) (commercially dehydrated base from Oxoid, Thermo Scientific, Basingstoke, United Kingdom) (*v*/*v*) at −80 °C until additional testing was performed. To perform further testing, the strains were subcultured on Columbia agar supplemented with 5% sheep blood (COS Agar) (bioMérieux, Marcy l’Etoile, France), incubated at 36 ± 1 °C in an atmosphere enriched with 5–10% CO_2_ for 18–24 h. Subsequently, a single colony was picked, subcultured on COS Agar, and incubated for another 18–24 h under the same conditions. 

### 4.2. Antimicrobial Susceptibility Testing (AST)

The minimal inhibitory concentration (MIC) was determined on a standard 0.5 McFarland bacterial suspension in a 0.85% saline solution using different susceptibility testing methods. The MIC is reported either in mg/L or μg/mL, and strains were classified as susceptible or resistant according to EUCAST breakpoints. 

#### 4.2.1. Broth Microdilution

Broth microdilution (BMD) was performed according to the method recommended by the European Committee on Antimicrobial Susceptibility Testing (EUCAST) (ISO 20776-1) and used as the reference method for antimicrobial susceptibility testing (AST) of rapidly growing aerobic bacteria. Both antibiotics used in this assay, vancomycin (V2002-100MG) and teicoplanin (T0578-100MG) (Sigma-Aldrich, Darmstadt, Germany), were resuspended in water at a concentration of 5120 mg/L (stock solution) and kept in aliquots at −80 °C until use. Ready-to-use antimicrobial solutions were freshly prepared from the stock solutions on the day of the assay using the Mueller–Hinton Broth (MHB). For the assay, 100 µL MHB was added in each well of a 96-well flat bottom plate. Then, 100 µL antibiotic with the highest concentration (32 mg/L) was added to the wells of the first column using a multichannel pipette, and mixed (pipetted 5 times), thus achieving a final concentration of 16 mg/L in the first dilution (wells A1-H1). Afterward, 100 µL suspension was transferred to the corresponding well in the second column. This process was repeated up to the 10th column, from which 100 µL were discarded. As a result, a serial twofold dilution was generated to a final concentration of 0.03 mg/L. In addition to the ten antibiotic concentrations columns, growth/positive control (column 11—MHB and bacterial inoculum without antibiotic) and negative control (column 12—only MHB) were tested. To all the wells other than the negative control column, 10 µL of the standard bacterial inoculum (5 × 10^5^ colony forming units/mL (CFU/mL)) was added. To obtain a standard inoculum, each strain was resuspended in 0.85% saline to a 0.5 on the McFarland scale (McF) (1–2 × 10^8^ CFU/mL), followed by a 1:20 dilution (5 × 10^6^ CFU/mL). The 96-well-plate was sealed and incubated for 24 h at 36 ± 1 °C air (according to ISO 20776-1), and the OD was then measured at 620 nm with a Sunrise TW absorbance reader (Tecan Trading AG, Männedorf, Switzerland). An absorbance of >0.5 was considered positive for bacterial growth. 

A second BMD assay was performed under similar conditions but with a higher bacterial inoculum (hBMD). For the bacterial inoculum, the strains were resuspended in 0.85% saline to a 0.5 McF, diluted 1:2 (5 × 10^7^ CFU/mL), and 10 µL added to the well to a final concentration of 5 × 10^6^ CFU/mL. The plates were sealed and incubated for 18 ± 2 h, and OD was measured. 

#### 4.2.2. Agar Dilution

For agar dilution (AD) assay, the Mueller–Hinton agar (dehydrated base from Oxoid, Thermo Scientific, Basingstoke, United Kingdom) was autoclaved and cooled to 45–50 °C and adjusted to a 7.3 pH, and teicoplanin from the stock solution was added to final concentrations of 0.25, 0.5, 1, 2, 4, and 8 mg/L. Additionally, drug-free plates were prepared and used for growth control. The prepared plates were kept wrapped at 4 °C and brought to room temperature before being subjected to previously described procedures [33,35,36]. Briefly, a 0.5 McF (1–2 × 10^8^ CFU/mL) standard bacterial suspension was serially diluted 1:10 to 10^3^ CFU/mL, and 10 µL from each dilution was transferred to the plates and incubated for 20–24 h and 48 h, after which the colonies were counted.

#### 4.2.3. Glycopeptide Antibiotic Susceptibility Testing (EUCAST) 

EUCAST endorses the use of standard gradient diffusion test (sGDT), macrodilution gradient test (MET), and screening agar as detection methods of glycopeptide non-susceptible *S. aureus* strains [37]. These assays have been recommended by EUCAST for *S. aureus* for research use only but have neither been suggested nor validated for CoNS. The obtained results are therefore not suitable for clinical interpretation. 

The teicoplanin standard gradient diffusion strip test (sGDT) was performed according to the manufacturer’s instruction using teicoplanin MIC test strips (range 0.016–256 µg/mL) (MTS; Liofilchem, Italy) [38] on a 0.5 McF standard bacterial inoculum on Mueller–Hinton agar (MHE) plates (BioMérieux, France). The MIC in mg/L was read after 16–20 h incubation, representing the point where the formed symmetrical ellipse met the strip. 

MET was performed according to EUCAST and the manufacturer’s instructions. Briefly, colonies from a 24 h old culture were resuspended in 2 mL 0.85% saline to McF 2 (heavier inoculum), streaked evenly on a brain–heart infusion (BHI) agar (Graso Biotech, Poland), and left to dry. Teicoplanin gradient strips were applied to the surface, incubated at 37 °C air, and read after 24 and 48 h. Not only was the value documented but also the presence of hazes, microcolonies, and isolated colonies. 

#### 4.2.4. Screening Agar

For the agar screening method, in-house Mueller–Hinton agar plates with and without 5 mg/L teicoplanin were produced and used based on the previously described protocol [39]. Briefly, colonies were suspended in 0.85% saline to an McF 0.5 and McF 2.0, and 10 µL of each inoculum were evenly distributed on the surface of the agar, incubated at 37 °C in air, and the growth was assessed after 24 and 48 h. 

#### 4.2.5. Disc Diffusion

Disc diffusion was performed, even though this approach is no longer EUCAST-recommended. CLSI version 2012 released breakpoints for disc diffusion warning indicating that it is unknown if the method can discriminate between susceptible and resistant strains to teicoplanin. For disc diffusion, the bacterial inoculum was evenly distributed on MHE plates (bioMérieux, Marcy l’Etoile, France), teicoplanin 30 mg discs (Liofilchem, Italy) placed on the surface, and incubated at 36 ± 1 °C in air. The inhibition zone was read after 24 h and interpreted according to the Clinical and Laboratory Standards Institute (CLSI). 

#### 4.2.6. Quality Controls

All the performed tests included negative and positive controls. *S. aureus* ATCC 29213 (teicoplanin reference range 0.25–1 mg/L, vancomycin reference range 0.5–2 mg/L) was included as a positive control (quality controls; QC strains) in all the assays under the same conditions as the CoNS strains [40]. The test results were considered valid only when the QC strain was tested within the EUCAST-given ranges. The AD assay included three additional strains as controls: *Enterococcus faecalis* ATCC 29212 (teicoplanin reference range 0.25–1 mg/L, vancomycin reference range 1–4 mg/L); the vancomycin-resistant *S. aureus* (VRSA) strain Mu50 (ATCC 700699); and Mu3 (ATCC 700698), a methicillin-resistant *S. aureus* (MRSA) strain with heterogeneous resistance to vancomycin. 

### 4.3. EUCAST Rules, Results Interpretation, and Data Analysis

All the AST results, except disc diffusion, were interpreted according to EUCAST breakpoints [41] and assigned to susceptible (MIC ≤ 4 mg/L) or resistant (MIC > 4 mg/L). The MIC values were reported in serial 1:2 dilutions and intermediate values as the next higher MIC. CLSI criteria were used to assess the results of disc diffusion and sBMD. According to CLSI, the strains were susceptible at MIC ≤ 8 mg/L, with zone diameter ≥14 mm; intermediate at MIC 16 mg/L, with zone diameter 11–13 mm; or resistant at MIC ≥ 32 mg/L, with zone diameter ≤ 10 mm [42].

Data were analyzed by comparing the measured MIC values and the corresponding interpretation generated using Vitek-2, hBMD, AD-24 h, AD-48 h, sGDT, MET, and screening agar with those from sBMD, the EUCAST recommended reference method. A very major error (vME) was defined as a false-susceptible result, whereas a major error (ME) was considered a false-resistant result compared with the results of sBMD. An essential agreement (EA) was considered when the MICs fell within the 1 log_2_ dilution of the MIC determined using sBMD, while categorical agreement (CA) was assigned to the isolate rated with the same interpretation category results (S/R) as sBMD. Acceptable performance for a method was defined as a percentage ≥90% for EA, CA, and ≤3% for vME or ME [31].

## Figures and Tables

**Table 1 antibiotics-12-00611-t001:** Annual resistance statistics for CoNS for the entire clinic between 2015 and 2020, susceptible strains in percentage (%).

Year	Total	OXA	GEN	LEV	SXT	ERN	CLI	VAN	TEI	LIN	TIG	FOS	FUS	RIF	TET	DAP
2015	650	36	57	44	72	30	47	100	**68**	100	99	56	61	93	55	100
2016	669	31	54	42	72	28	44	100	**71**	100	100	57	-	92	-	99
2017	759	32	58	45	72	31	43	100	**74**	100	100	51	-	93	-	99
2018	619	39	64	49	73	33	50	100	**69**	100	100	59	-	92	-	100
2019	562	36	63	54	71	34	50	100	**84**	99	100	56	-	92	-	99
2020	497	37	66	54	70	36	52	100	**80**	99	100	61	-	94	-	98

OXA (oxacillin), GEN (gentamicin), LEV (levofloxacin), SXT (trimethoprim–sulfamethoxazole), ERN (erythromycin), CLI (clindamycin), VAN (vancomycin), TEI (teicoplanin), LIN (linezolid), TIG (tigecycline), FOS (fosfomycin), FUS (fusidic acid), RIF (rifampicin), TET (tetracycline), DAP (daptomycin).

**Table 2 antibiotics-12-00611-t002:** Antimicrobial resistance routinely performed using Vitek-2 *.

Vitek-2	EUCAST							
	Susceptible (S) ≤ 4 mg/L (%)				Resistant > 4 mg/L (%)			Total
MIC Teicoplanin mg/L	≤0.5	1	2	4	8	16 *	32 *	
*S. epidermidis*	15 (9.3)	3 (1.8)	24 (14.8)	46 (28.4)	44 (27.2)	24 (14.8)	1 (0.6)	157
*S. haemolyticus*	-	-	-	-	1 (0.6)	1 (0.6)	-	2
*S. hominis*	-	-	-	-	1 (0.6)	2 (1.2)	-	3
Total %	88 (54.3)				74 (45.7)			162

* According to CLSI: strains with a MIC of 16 mg/L would be assigned to intermediate strains and the strain with a MIC of 32 mg/L would be resistant.

**Table 3 antibiotics-12-00611-t003:** Teicoplanin MIC distribution with Vitek-2.

Tei	AB	OXA	GEN	LEV	SXT	ERN	CLI	VAN	LIN	TIG	FOS	FUS	RIF	TET	DAP
S (88)	R	62	37	42	23	59	41	-	-	-	17	42	5	57	1
	S	26	51	46	65	29	57	88	88	88	71	46	83	31	87
R (74)	R	66	38	58	25	57	53	-	-	-	17	35	3	29	-
	S	8	36	16	49	17	21	74	74	73 **	56 **	28 **	71	45	73 **
Total															
	R	128	75	100	48	116	94	-	-	-	34	77	8	86	1
	%	79	46.3	61.7	29.6	71.6	58.0	-	-	-	21.0	47.5	4.9	53.1	0.6
	S	34	87	62	114	46	68	162	161	161	127	84	154	76	160
	%	21	53.7	38.3	70.4	28.4	42.0	100	99.4	99.4	78.4	51.8	95.1	46.9	98.7

AB (antibiotic), OXA (oxacillin), GEN (gentamicin), LEV (levofloxacin), SXT (trimethoprim–sulfamethoxazole), ERN (erythromycin), CLI (clindamycin), VAN (vancomycin), TEI (teicoplanin), LIN (linezolid), TIG (tigecycline), FOS (fosfomycin), FUS (fusidic acid), RIF (rifampicin), TET (tetracycline), DAP (daptomycin). ** For one strain, TIG, FOS, FUS, and DAP were not tested.

**Table 4 antibiotics-12-00611-t004:** sBMD MICs.

sBMD	EUCAST							
	Susceptible (S) ≤ 4 mg/L (%)				Resistant > 4 mg/L (%)			Total
MIC Teicoplanin mg/L	≤0.5	1	2	4	8	16	>16	
*S. epidermidis*	15	29	48	53	12	-	-	157
*S. haemolyticus*	-	-	-	-	-	1 *	1 *	2
*S. hominis*	-	-	-	-	2	-	-	3
	9.3	17.9	29.6	33.3	8.6	0.6	0.6	162

* According to CLSI: the strain with a MIC of 16 mg/L would be assigned to intermediate and the strain with a MIC of >16 would be resistant.

**Table 5 antibiotics-12-00611-t005:** The minimum inhibitory concentration of the staphylococci strains using hBMD.

hBMD	EUCAST							
	Susceptible (S) ≤ 4 mg/L (%)				Resistant > 4 mg/L (%)			Total
MIC Teicoplanin mg/L	≤0.5	1	2	4	8	16	>16	
*S. epidermidis*	11	16	35	45	44	5 *	1 *	157
*S. haemolyticus*	-	-	-	-	2	-	-	2
*S. hominis*	-	-	-	-	1	-	-	3
	6.8	9.9	21.6	29.0	29.0	3.1	0.6	162

* According to CLSI: the strain with a MIC of 16 mg/L would be assigned to intermediate and the strain with a MIC of > 16 would be resistant.

**Table 6 antibiotics-12-00611-t006:** MIC distribution using sBMD vs. Vitek-2 and the respective EA, CA, and ME.

EUCAST Category	sBMD Teicoplanin mg/L	sBMD No.	MIC by Vitek-2, (No.)							
			Susceptible				Resistant			
			0.5	1	2	4	8	16	32	
Resistant (R)	32	-	-	-	-	-	-	-	-	EA 81 (50) CA 106 (65.4) ME 59 (36.4)
	16	2	-	-	-	-	2	-	-	
	8	11	-	-	-	-	3	7	1	
Total R		13 (8%)	-				13			
Susceptible (S)	4	40	-	-	4	6	18	12	-	
	2	49	2	2	4	20	18	3	-	
	1	22	2	1	6	7	3	3	-	
	0.5	25	5	-	8	11	-	1	-	
	<0.5	11	6	-	3	1	1	-	-	
Total S		147 (90.7%)	88 (54.3)				59 (36.4)			

Notably, 2 strains of 162 were not tested due to lack of growth. EA, essential agreement; CA, categorical agreement; ME, major error.

**Table 7 antibiotics-12-00611-t007:** MICs using agar dilution after 24 h and 48 h incubation.

Agar Dilution (AD) *	EUCAST									
	Susceptible (S) ≤ 4 mg/L								Resistant > 4 mg/L	
Incubation	24 h	48 h	24 h	48 h	24 h	48 h	24 h	48 h	24 h	48 h
Teicoplanin (mg/L)	0.5		1		2		4		≥8	≥8
*S. epidermidis*	1	1	16	6	61	46	48	44	31	59
*S. haemolyticus*	-	-	-	-	-	-	-	-	-	2
*S. hominis*	-	-	-	-	-	-	-	-	2	3

* One strain remained without growth.

**Table 8 antibiotics-12-00611-t008:** Results of AD-24 h vs. other AST assay methods.

AD-24 h	sBMD	AD-48 h	Vitek-2	sGDT	Screening McF 0.5	Screening McF 2	Disc Diffusion (CLSI)
128 S	125 S	97 S	80 S	125 S	80 pos	114 pos	126 S
	3 R	31 R	48 R	1 R	44 neg	12 neg	-
	-	-	-	2 NE	4 NE	2 NE	2 NE
33 R	20 S	-	7 S	31 S	33 pos	33 pos	31 S
	13 R	33 R	26 R	2 R	-	-	2 I
	-	-	-	-	-	-	
1 NG	1 S	NG	1 S	NE	-	-	NG

S, susceptible; R, resistant; NE, not evaluable; NG, no growth; pos, positive; neg, negative; I, intermediate according to CLSI.

**Table 9 antibiotics-12-00611-t009:** Results of AD-48 h vs. other AST methods.

AD-24 h	sBMD	AD-48 h	Vitek-2	sGDT	Screening McF 0.5	Screening McF 2	Disc Diffusion (CLSI)
97 S	95 S	97 S	74 S	96 S	50 pos	84 pos	96 S
	2 R	-	23 R		44 neg	12 neg	-
	-	-	-	1 NE *	3 NE	1 NE	1 NE
64 R	50 S	31 S	13 S	60 S	63 pos	63 pos	60 S
	14 R	33 R	51 R	3 R	-	-	2 I
	-	-	-	1 NG	1 NE-	NE	2 NE
1 NG	1 S	NG	1 S	NE	-	-	NG

* S, susceptible; R, resistant; NE, not evaluable; NG, no growth; pos, positive; neg, negative; I, intermediate according to CLSI.

**Table 10 antibiotics-12-00611-t010:** Susceptibility results using sGDT.

sGDT	EUCAST					
	Susceptible (S) ≤ 4 mg/L (%)				Resistant > 4 mg/L (%)	
MIC Teicoplanin mg/L	≤0.5	1	2	4	8	16
*S. epidermidis*	23	81	42	8	-	-
*S. haemolyticus*	-	-	-	-	2	-
*S. hominis*	-	-	2	-	-	1
% *	14.2	50	27.2	4.9	1.2	0.6

* Two strains display no growth, and for one strain, the MIC could not be read (1.8%).

**Table 11 antibiotics-12-00611-t011:** Comparison of strains with high MET values (≥8 mg/L) with other AST assays.

No.	MET	sBMD	hBMD	Vitek-2	sGDT	AD-24 h	AD-48 h	Screening McF 0.5	Disc Diffusion	Material	Strain ID
71	8	R	R	R	R	S	R	pos	S	BC	*S. haemolyticus*
72	12	R	R	R	R	R	R	pos	I	BC	*S. hominis*

sBMD, standard broth microdilution; hBMD, high-broth microdilution; sGDT, standard gradient strip; BC, blood culture; AD, agar dilution; S, susceptible; R, resistant; I, intermediate according to CLSI; pos, positive.

**Table 12 antibiotics-12-00611-t012:** Teicoplanin susceptibility tested via AST and the EA, CA, vME, and ME yielded when compared with sBMD.

Method	Strain	No. % Isolates		EA	CA	vME	ME
		Susceptible	Resistant				
		≤4	>4				
sBMD	All strains	146 (90.1)	16 (9.9)				
	*S. epidermidis*	145 (89.5)	12 (7.4)				
	*S. haemolyticus*	-	2 (1.2)				
	*S. hominis*	1 (0.6)	2 (1.2)				
hBMD	All strains	109 (67.3)	53 (32.7)	137 (84.6)	121 (74.7)	2 (1.2)	39 (24.1)
	*S. epidermidis*	107 (66)	50 (30.9)	132 (81.5)	117 (72.2)	1 (0.6)	39 (24.1)
	*S. haemolyticus*	-	2 (1.2)	2 (1.2)	2 (1.2)	-	-
	*S. hominis*	2 (1.2)	1 (0.6)	3 (1.8)	2 (1.2)	1 (0.6)	-
Vitek-2	All strains	88 (54.3)	74 (45.7)	103 (63.6)	94 (58.0)	5 (3.1)	63 (38.9)
	*S. epidermidis*	88 (54.3)	69 (42.6)	99 (61.1)	90 (55.6)	5 (3.1)	62 (38.2)
	*S. haemolyticus*	-	2 (1.2)	2 (1.2)	2 (1.2)	-	-
	*S. hominis*	-	3 (1.8)	2 (1.2)	2 (1.2)	-	1 (0.6)
AD-24 h ^1^	All strains	128 (79)	33 (20.4)	146 (90.1)	138 (85.2)	3 (1.8)	20 (12.4)
	*S. epidermidis*	125 (77.2)	31 (19.1)	142 (87.7)	134 (82.7)	2 (1.2)	20 (12.4)
	*S. haemolyticus*	1 (0.6)	1 (0.6)	1 (0.6)	1 (0.6)	1 (0.6)	-
	*S. hominis*	1 (0.6)	2 (1.2)	3 (1.8)	3 (1.8)	-	-
AD-48 h ^1^	All strains	97 (59.9)	64 (39.5)	132 (81.5)	109 (67.3)	2 (1.2)	50 (30.9)
	*S. epidermidis*	97 (59.9)	59 (36.4)	127 (78.4)	105 (64.8)	2 (1.2)	49 (30.2)
	*S. haemolyticus*	-	2 (1.2)	2 (1.2)	2 (1.2)	-	-
	*S. hominis*	-	3 (1.8)	3 (1.8)	2 (1.2)	-	1 (0.6)
sGDT ^2^	All strains	156 (96.3)	3 (1.8)	118 (72.8)	146 (90.1)	13 (8.0)	-
	*S. epidermidis*	154 (95.1)	-	114 (70.4)	142 (87.7)	12 (7.4)	-
	*S. haemolyticus*	-	2 (1.2)	2 (1.2)	2 (1.2)	-	-
	*S. hominis*	2 (1.2)	1 (0.6)	2 (1.2)	2 (1.2)	1 (0.6)	-

Strains without growth: ^1^ one strain and ^2^ three strains.

## Data Availability

The data presented in this study are presented in the tables, no additional data was generated.

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
