# Peer review of "Teicoplanin-Resistant Coagulase-Negative Staphylococci: Do the Current Susceptibility Testing Methods Reliably Detect This Elusive Phenotype?"

_antibiotics, 2023, doi:10.3390/antibiotics12030611_

Round 1

Reviewer 1 Report

The manuscript investigates various testing methods for the detection of teicoplanin resistance of Coagulase negative staphylococci (CoNS) strains. They used various techniques i.e., standard and high bacterial inoculum 16 broth microdilution (sBMD; hBMD), agar dilution (AD), standard and 17 macro gradient diffusion strip (sGDT, MET), screening agar and disc diffusion. The results show that none of the techniques is conclusive and can be used as screening method for the identification of  Staphylococci (CoNS) strains. The experiments are well performed and study sheds light on the drawbacks of current standard techniques.

Minor comments

1.    Section 2.1: Species name should be italics. (Change at others places also)

2.    Abstract needs to be more concise highlighting the main objectives and important findings, rather than just dividing it into different subsections.

3.    I would suggest authors to simplify their tables i.e., Table 6. It is difficult to follow the table with so many sub columns and rows.

4.    There are many grammatical mistakes which needs to be addressed. Follow the same pattern for formatting.

Author Response

  1. Section 2.1: Species name should be italics. (Change at others places also)

All species names have been corrected.

  1. Abstract needs to be more concise highlighting the main objectives and important findings, rather than just dividing it into different subsections.

Abstract modified as suggested (highlighted in grey).

  1. I would suggest authors to simplify their tables i.e., Table 6. It is difficult to follow the table with so many sub columns and rows.

We have changed the more complex tables and tried to make them easier to read. Unfortunately, the data are complex and thus a presenting them in simple tables is not possible. We believe that the changes have been successful in making the tables more accessible

  1. There are many grammatical mistakes which needs to be addressed. Follow the same pattern for formatting.

We have scrutinized the manuscript very carefully and did indeed find a number of errors, which have been corrected (mostly typographical and formatting errors).

Reviewer 2 Report

Authors aimed to assess the robustness of the routinely employed susceptibility testing by comparing it with other available methods; and to propose a diagnostic algorithm to detect the teicoplanin resistance and heteroresistance. The work approaches an important issue and should be of interest  for the readers of the Journal.

I am listing below points that  should be addressed by Authors:

Authors: “The Coagulase negative staphylococci (CoNS) include a large number of different Staphylococcus species and are part of the skin and mucous membrane commensal microbiota. In certain circumstances (interference in skin health, ecology, structure, and im

mune system) they may cause opportunistic local or systemic infections. The advances of modern medicine have led to an increase in the role of CoNS among immunocompromised, critically ill, long term hospitalized and patients with implanted medical devices.

> Provide references.

Authors: “CoNS strains have been reported to play a significant role not only among device- 

associated infections (intravascular catheters, cerebrospinal fluid shunts, prosthetic joint, vascular grafts, peritoneal-dialysis catheters), but also in cases of osteomyelitis, infective 39

endocarditis, surgical site infections and infections in neonates.”

>Provide references

Authors: “Van Epps et al. showed that 50-70% of the healthcare associated infections in USA are a consequence of a….”

> A recent reference is necessary.

Authors: “Teicoplanin shows less oto- and nephrotoxity than vancomycin and is therefore a therapeutic alternative to vancomycin.

> Provide references

Authors: “assess the robustness of the routinely employed susceptibility testing by 68

comparing it with other available methods;

> Authors should highlighted AST methods for teicoplanin in the Introduction section. 

Table 1. Table 1: Annual resistance statistics for CoNS for the entire clinic between 2015 and 2020, susceptible strains in percentage (%)

> Authors could highlight teicoplanin-resistance results on Table 1.

About AST methods:

> Authors should to analyze  categorical agreement (CA), very major errors (VME) and major errors (ME).

Author Response

  1. Authors: “The Coagulase negative staphylococci (CoNS) include a large number of different Staphylococcus species and are part of the skin and mucous membrane commensal microbiota. In certain circumstances (interference in skin health, ecology, structure, and immune system) they may cause opportunistic local or systemic infections. The advances of modern medicine have led to an increase in the role of CoNS among immunocompromised, critically ill, long term hospitalized and patients with implanted medical devices. > Provide references.

Available reference has been re-evaluate and inserted accordingly. The paper of Kloos et al. was added to the references because it represents a critical moment when CoNS was credited as an emerging opportunistic pathogen. Line 38-39

  1. Authors: “CoNS strains have been reported to play a significant role not only among device associated infections (intravascular catheters, cerebrospinal fluid shunts, prosthetic joint, vascular grafts, peritoneal-dialysis catheters), but also in cases of osteomyelitis, infective endocarditis, surgical site infections and infections in neonates.” >Provide references

The reference has been re-evaluated and inserted accordingly (Line 43)

  1. Authors: “Van Epps et al. showed that 50-70% of the healthcare associated infections in USA are a consequence of a….”> A recent reference is necessary.

We have evaluated the reviewer recommendation, but we consider that the paper of van Epps et al. summarizes the rate of infection for the most common implantable medical devices better than the more recent literature. Other recent reference are available e.g Boriani et al. (Infections associated with cardiac electronic implantable devices: economic perspectives and im-pact of the TYRX™ antibacterial envelope), however the only one indwelling device does not reflect the message that we would like to convey. Line 43

  1. Authors: “Teicoplanin shows less oto- and nephrotoxity than vancomycin and is therefore a therapeutic alternative to vancomycin. > Provide references

We have re-examined these aspects carefully and after thorough search we have corrected it and added the appropriate references. Lines 63-66

  1. Authors: “assess the robustness of the routinely employed susceptibility testing by comparing it with other available methods;> Authors should highlighted AST methods for teicoplanin in the Introduction section.

The methods for AST have been emphasised in the introduction lines 56-57

  1. Table 1. Table 1: Annual resistance statistics for CoNS for the entire clinic between 2015 and 2020, susceptible strains in percentage (%) > Authors could highlight teicoplanin-resistance results on Table 1.

Teicoplanin-susceptibility data has been changed to bold in the table 1.

  1. About AST methods: > Authors should to analyze categorical agreement (CA), very major errors (VME) and major errors (ME).

The breakdown of the Errors (CA,vME and ME) are depicted in the table 12, we have added a short a conclusion in text (line 268-269) to extrapolate on this.

Reviewer 3 Report

Dear Authors, 

I reviewed your manuscript entitled Teicoplanin-resistant coagulase-negative Staphylococci: current testing methods are discordant in the detection of this elusive phenotype. The article assesses the robustness of the routinely employed susceptibility testing by comparing it with other available methods, the main purpose being to develop a diagnostic algorithm in order to detect the teicoplanin resistance and heteroresistance thus avoiding labor-intense population analysis method.

The article is well-written and scientifically sound. 

I suggest some minor revisions:

1. I suggest the title to be reformulated as an inquiry, not as a conclusion. 

2. From the abstract, please retract the names of the categories required to be included (e.g. materials and methods). Just write it cursively as a short presentation of the article, including a conclusion. 

3. Please add a section 5 for conclusions at the end of the manuscript. 

Good luck! 

Author Response

  1. I suggest the title to be reformulated as an inquiry, not as a conclusion.

The title has been reformulated as a question (this was in fact one of our choices before submission)

  1. From the abstract, please retract the names of the categories required to be included (e.g. materials and methods). Just write it cursively as a short presentation of the article, including a conclusion.

The abstract has been modified accordingly and a conclusion included.

  1. Please add a section 5 for conclusions at the end of the manuscript

As suggested by the reviewer the conclusions have been added at the end of discussion section lines 301-309.

Reviewer 4 Report

The manuscript entitled "Teicoplanin-resistant coagulase-negative Staphylococci: current testing methods are discordant in the detection of this elusive phenotype" publishes the results of a study of staphylococcus resistance to teicoplanin conducted by the authors using a variety of methods. Considering that the problem of bacterial resistance is acute, the article is of scientific interest. Quite interesting work has been carried out, since laboratory testing of existing and massively used therapeutic techniques can show both their effectiveness and reveal inaccuracy and unreliability. The results of the work should be considered as a statement of the question that needs to be solved in the future. The text is written correctly, logically and consistently. Rather monotonous tables could be supplemented with a graphic abstract in the form of a diagram or figure. Conclusions should preferably be written separately. It is recommended to accept the article after minor changes.

1) Tables 3, 6 and 12 are placed on two pages, each. It is advisable to put them each on one page.

2) Numbering error. Paragraphs 4.2.4 and 4.2.5 are missing, but paragraph 4.2.6 is written twice.

3) Line 413. It is advisable not to separate the chapter title from the text.

4) Lines 498 and 513. Extra characters that were not deleted during editing.

5) It is possible that the names of subsections may not be written in the Abstract text: Background, Material and methods, Results and Discussion.

6) Lines 209-213. Extra lines.

7) Abstract is overloaded with numbers that make it difficult to perceive the material. It is advisable to use more general formulations and conclusions. The reader can always refer to the Results chapter for figures and specifics.

8) A short chapter of Conclusions is missing. Considering that various methods have yielded contradictory results, it is necessary to single out the most important points from the entire volume of data and present them theses. In addition, recommendations for assessing the resistance of staphylococcus to teicoplanin in clinical trials are desirable.

Author Response

  1. Rather monotonous tables could be supplemented with a graphic abstract in the form of a diagram or figure.

The suggestion of the reviewer is valid; however, it is very difficult to present the data in a graphical manner and does not present the data in a very much more easily readable fashion.

  1. Conclusions should preferably be written separately

Conclusions has been added.

  1. Tables 3, 6 and 12 are placed on two pages, each. It is advisable to put them each on one page.

This is due to formatting differences and will presumably be changed at the typesetting stage. We have inserted a page-break.

  1. Numbering error. Paragraphs 4.2.4 and 4.2.5 are missing, but paragraph 4.2.6 is written twice.

The numbering has been corrected accordingly.

  1. Line 413. It is advisable not to separate the chapter title from the text

This should be corrected at the typesetting stage. We have inserted a page-break.

  1. Lines 498 and 513. Extra characters that were not deleted during editing.

The extra characters have been deleted.

  1. It is possible that the names of subsections may not be written in the Abstract text: Background, Material and methods, Results and Discussion.

Corrected

  1. Lines 209-213. Extra lines

Corrected

  1. Abstract is overloaded with numbers that make it difficult to perceive the material. It is advisable to use more general formulations and conclusions. The reader can always refer to the Results chapter for figures and specifics.

The abstract has been modified accordingly.

  1. A short chapter of Conclusions is missing. Considering that various methods have yielded contradictory results, it is necessary to single out the most important points from the entire volume of data and present them theses. In addition, recommendations for assessing the resistance of staphylococcus to teicoplanin in clinical trials are desirable

These recommendations have been included and the important data emphasised in the conclusions.